# Artificial intelligence tool for the study of COVID-19 microdroplet spread across the human diameter and airborne space

Hesham H. Alsaadi[1☯], Monther Aldwairi[1☯¤]*, Faten Yasin[2☯], Sandra C. P. Cachinho[3☯], Abdullah Hussein[4☯]

**1** College of Technological Innovation, Zayed University, Abu Dhabi, UAE, **2** Department of Molecular and Clinical Cancer Medicine, University of Liverpool, Liverpool, United Kindom, **3** Cell Sorting and Isolation Facility, Research Technology Building, University of Liverpool, Liverpool, United Kindom, **4** Rochester Institute of Technology, Dubai, UAE

☯ These authors contributed equally to this work.
¤ Current address: Department of Network Engineering and Security, Jordan University of Science and Technology, Irbid, Jordan
* monther.aldwairi@zu.ac.ae

**Data Availability Statement:** COVNET45 is currently hosted in AWS and can be integrated with the AMAZON AI machine learning platform Tool,

## Abstract

The 2019 novel coronavirus (SARS-CoV-2 / COVID-19), with a point of origin in Wuhan, China, has spread rapidly all over the world. It turned into a raging pandemic wrecking havoc on health care facilities, world economy and affecting everyone's life to date. With every new variant, rate of transmission, spread of infections and the number of cases continues to rise at an international level and scale. There are limited reliable researches that study microdroplets spread and transmissions from human sneeze or cough in the airborne space. In this paper, we propose an intelligent technique to visualize, detect, measure the distance of spread in a real-world settings of microdroplet transmissions in airborne space, called "COVNET45". In this paper, we investigate the microdroplet transmission and validate the measurements accuracy compared to published researches, by examining several microscopic and visual images taken to investigate the novel coronavirus (SARS-CoV-2 / COVID-19). The ultimate contribution is to calculate the spread of the microdroplets, measure it precisely and provide a graphical presentation. Additionally, the work employs machine learning and five algorithms for image optimization, detection and measurement.

## Introduction

The 2019 novel coronavirus (SARS-COV2 / COVID-19) is an infectious respiratory disease that shares the same means and mechanisms of transmission as influenza [1]. Infectious disease transmitted through respiratory secretion can spread via:

- Droplets—large particles ($>5$ μm) that travel under 1m.

- Aerosols—smaller particles ($<5$ μm) that travel over 1m.

Data and code will be shared via https://github.com/HeshamAlsaadi.

**Funding:** MA Research Incentive Fund Grant #R20089 Zayed University Research Office www.zu.ac.ae The funders had no role in study design, data collection and analysis, decision to publish, or preparation of the manuscript.

**Competing interests:** The authors have declared that no competing interests exist.

• Contact—with objects that have been infected with droplets and then touching the eyes, nose, or mouth [2].

The airborne transmission does not require direct contact between infected and susceptible individuals; therefore the virus causing COVID-19 might spread via droplets, e.g. when a person coughs. One can inhale microscopic aerosol particles consisting of the reliable residual components of evaporated respiratory droplets, which are tiny enough to remain airborne for hours. The aerosolized SARS-CoV-2 can remain viable and infectious in aerosols for hours and on certain surfaces up to days. But this begs the question: how do asymptomatic infected individuals generate aerosols? The normal acts of breathing or speaking could emit large quantities of aerosol particles. They are of approximately 1 μm to 500 μm in diameter but are big enough to carry viruses. COVID-19 is a highly transmissible infectious respiratory illness that shares the routes and means of transmission with influenza. The largest size of microdroplets >5 μm can travel under 1m, while the smaller microdroplet particles <5 μm in the area can go over 1m [3].

Furthermore, some individuals are "super speech emitters," having the ability to emit more aerosol particles than others. For example, a 10 minutes conversation with an infected, asymptomatic super emitter talking in an average volume, could generate an invisible "cloud" of approximately 6,000 aerosol particles that could potentially be inhaled by the other party [4–6]. Therefore, vital to study and understand the dynamics of the spread of cough and breathing particles of different sizes.

Gralton summarized the size of coughed particles from a large number of studies and concluded that the size of cough-generated particles ranged from 0.1–100 μm [7]. To study the stability of COVID-19, some research groups employed artificially generated and aged aerosols using a nebulizer, and suspended it in the air with a Goldberg drum. The viruses, in different environmental conditions, would remain viable in aerosols for 3 hours throughout the experiment; however, the virus titer was significantly reduced after 3 hours [8].

At the present time, there are vaccines available and administered to prevent the serious side effects of the virus, but they do not stop the spread [9]. Considering that it is still a highly contagious virus, mask mandates in closed areas remain the main measure to prevent any droplet inhalation. Therefore, the best way of preventing spread is by avoiding exposure, and the mandate of personal protective measures, which remain vital to mitigate and control the pandemic. Consequently the Centre for Disease Control and Prevention (CDC) and The World Health Organization (WHO) recommend that people avoid touching their faces [10]. The recommendation is to wear masks and face shields to minimize the risk of transmission [11, 12].

However, wearing masks may create a false sense of security, therefore the subject should not neglect all the other essential preventative measures: hand hygiene, social distance, respiratory etiquette, and self-isolation if in close contact or exhibiting any symptoms [12]. Different masks' types have different protection effectiveness and different breathabilities. For example, N95 and surgical masks, have a higher protection with more than 90% efficiency against particles of 0.3-4.5 μm and close to 100% for particles larger than 4.5 μm. It was shown that wearing a surgical face mask could prevent or reduce transmission of human coronaviruses and influenza viruses from symptomatic individuals. However, as the pandemic continues to claim lives, and calls for relaxing mask mandates, such as lifting the restrictions in outdoor and open spaces, the need for masks indoors remains a must [13].

In this paper, we propose an intelligent technique to visualize, detect, and measure the distance of the spread of microdroplet transmissions in airborne space in a real-world setting. The article proposes and evaluates a complete framework and software that uses five

techniques as well as machine learning for image optimization, object detection, visualization and measurement. The main aim is to develop a novel image optimization technique for measuring the distance of microdroplets spread and transmissions in airborne space.

The main contributions of this work are:

- Generate high accuracy measurement by optimizing the collected images and their characterization.

- Combine the strong capabilities of accurately measuring spread with the use of different scale systems: nominal, ordinal, interval and ratio.

- Provide a generic approach that is not limited to small scale microdroplets transmissions, but also able to measure the diameter of spread in any space using 2D graphical images with the capabilities of analyzing, optimizing and detecting various semantic marks.

- Provide a versatile tool capable of analyzing different images formats, which may enable researcher to validate previous research results and compare to related work

- Latest AES-256 encryption is used to protect the hosted images in AWS cloud to ensure added privacy

The tool has been made publicly available to other researchers along with all the images and full detailed technical guide [14, 15]. The rest of the paper is organized as follows. Section 2 discusses the existing related work and other proposed methods for measurement systems in the literature. In Section 3, we describe the methodology, proposed approach and illustrate the inner workings of the COVNET45 tool. In Section 4, we show the implementation details and experimental results, and discuss COVNET45 limitations. Section 5 concludes the paper.

## Related work

Several studies have indicated that identifying the leading causes of the spread of COVID-19 may be through the transferal of infinitely small particles through the air. Proposed research used a background oriented Schlieren technique to investigate the airflow ejected by a person quietly or heavily breathing, and while coughing. They tested the effectiveness of different face covers including: FFP2 mask, FFP1 mask, a respirator, a surgical mask, a hand-made mask, and two types of face shields. They simulated an aerosol-generating procedure and demonstrated the extent of aerosol dispersion [16]. The study concluded that all face covers except the respirator, allow a reduction of the front flow through jet by more than 90%. Although the results sound promising the experimental setup does not reveal the absolute maximum distance that a virus-laden fluid particle can travel, nor how the concentration of these particles varies spatially and temporally [17].

Another proposed study described a solution for rapid detection of COVID-19, using genotypic testing for SARS-CoV-2 virus in nasopharyngeal. The study demonstrated fluorescent microscopy, and CT scan images of COVID-19 patients along with whole patient blood and platelet-poor plasma. The study concluded that micro clots can be detected in the native plasma of COVID-19 patient, and in particular that such clots are amyloid in nature as judged by a standard fluorogenic stain. Moreover, the study found that the plasma of COVID-19 patients carries a massive load of preformed amyloid clots. These clots imaged with TEG to provide a rapid, early detection test for clotting severity in such patients. One of the study limitations was the indication of the virus spread within the micrographic images of the plasma test [18].

Another study investigated the speech droplets generated by asymptomatic carriers of severe acute respiratory syndrome SARS-CoV-2). The research considered that it is likely the mode of disease transmission. Highly sensitive laser light scattering observations have revealed that loud speech can emit thousands of oral fluid droplets per second. In a closed, stagnant air environment, they disappear from the window of view within 8 to 14 min, which corresponds to droplet nuclei of ca. 4 μm diameter, or 12- to 21-μm droplets before dehydration. These observations confirm that there is a substantial probability that normal speaking causes airborne virus transmission in confined environments. The study cannot provide validation accuracy of the size and the diameter of the droplet using light scattering observation of airborne speech droplet nuclei [3].

A recent study demonstrated the potential of microdroplet infection transmission in movement, such as walking fast, running and cycling, can increase the infection spread to other individuals in public areas. The aerodynamics study investigated whether, a first-person moving nearby or a second person at 1.5 m distance or beyond, could cause droplet transfer. The study demonstrated a CFD simulation of droplet dispersion around two walkers with exhaling velocity of 2.5 m/s relative to the movement of the walker/runner, representing moderately deep breathing. Moreover, the research experiments illustrated water droplets presented as (saliva) released at a total flow rate of 1x10-14 mg/s with a Rosin-Rammler droplet distribution with a minimum diameter of 40 μm, an average diameter of 80 μm and a maximum diameter of 200 μm. The study concluded that for fast walking at 4 km/h the spread distance is about 5 m and for running at 14.4 km/h, the spread distance is about 10 m [19].

Similar research explained the role of aerosols in the transmission of COVID-19, the research described two possible modes of COVID-19 aerosols transmission: a) during a sneeze or a cough, "droplet sprays" of virus-laden respiratory tract fluid, typically greater than 5 μm in diameter, impact directly on a susceptible individual and b) alternatively, a susceptible person can inhale microscopic aerosol particles consisting of the residual solid components of evaporated respiratory droplets, which are tiny enough <5 μm) to remain airborne for hours [20].

Some studies used practical simulation to study the method of transmission by using robots to study the molecular splash of droplets spread to demonstrate the diffusion process indicating spread up to 8 feet of molecules through the air for individuals without the mask [21]. Moreover, the study used artificial materials of droplets simulated as real microdroplets from a human. Very concise recent studies have illustrated that the COVID-19 virus distribution process is one of the principal causes of widespread transmission through the air, as the amount of spread caused by an individual varies by the size of the droplets exhaled by the human body, which differs from sneezing or coughing.

Yamamoto et al. proposed a probabilistic mathematical model for the distribution of airborne and droplets to predict Covid-19 infection [22]. For the parameters needed they relied on known estimation methods, however no real measurement or images processing was performed. It is true they applied this prediction model in a real office where they measured $CO_2$ concentrations and the occupant positions, however there was no image analysis or measurement to study the spread of particles. Similarly, Oliveira et al. [23] presented a theoretical mathematical estimate of virus transmission to provide social distancing and ventilation measures, however none of that is comparable to actual microdroplet measurements capabilities of COVNET45.

More studies have dealt with the issue of airborne contamination paths, counting of pathogens, quantum concentration, carbon dioxide concentration and other measurements to give recommendations about ventilation and indoor capacity [24]. Additionally, there is a wealth of work in the area of Indoor air quality (IAQ) to use sensors and visualization in preventing

indoor spread of SARS-CoV2 [25]. We believe those studies could benefit from the measurement capabilities of droplet transmission provided by COVNET45.

The research studies, mentioned earlier, adopted different techniques for scaling and measuring microdroplets infection transmissions either by simulating graphical representation of microscopic images or by visualizing the spread using Computational Fluid Dynamics (CFD) examination. Many of the discussed studies lack the evidentiary tools to validate their droplets measurement results, this makes it more complicated to authenticate the results in every research investigating the virus droplets spread through patient cough or sneeze. Some studies also supported only specific data formats (CT scans, micrographic/microscopic images, CFD images, video graphics), and even when handling multiple formats, they cannot compare or validate results because of the restrictions presented by the investigation technology used. The fact that microdroplets patterns are ideally identifiable and recognizable makes it challenging to measure the precise size, even when using automated scaling systems implemented within the technology. Finally, extensive types of scales' formats for accurately measuring microdroplets or any microscopic details within the hierarchy is challenging, which leaves the door open to investigate more optimized tools to explore and authenticate studies about the infectious disease.

Next, we explain the proposed methodology and the structure of the proposed tool, COVNET45, which is optimized for droplets measurements using graphical user interface (GUI) representation that generates unique insights and accurate results.

## Methodology

COVNET45 uses images as a data source for processing and optimizing. In this section, first, we show the level relational paradigm of microscopy images. Second, we explain the process of optimizing microscopy images using multiple machine learning algorithms. Third, we use a scale calculation method to validate the results. Finally, we export and visualize the optimized solution.

### Objects, events and change management

In this section, we detail the process of managing microscopic images of COVID-19 virus using electron microscope device, how the COVID-19 virus is distributed and organized in each cell and broken into objects in a single microscopic image file. In COVNET45, we consider that a single spherical virus particle containing black dots of COVID-19 virus as an *object* in a single microscopic image, and the possible movements/diameter in size of a single spherical virus particle as *events*. Fig 1A and 1B show the relational paradigm ontology of objects, events and change, and how the basic framework processes a sample thin-section electron microscopic image of SARS-CoV-2, the causative agent of COVID-19. Spherical virus particles contain black dots, which are cross-sections through the viral nucleocapsid. In the cytoplasm of the infected cell, clusters of particles are found within the membrane-bound cisternae of the rough endoplasmic reticulum/Golgi area.

Fig 1B(I) shows a sample image of thin-section electron microscopic image of SARS-CoV-2, the causative agent of COVID-19. Spherical virus particles contain black dots, which are cross-sections through the viral nucleocapsid. In the cytoplasm of the infected cell, clusters of particles are found within the membrane-bound cisternae of the rough endoplasmic Reticulum/Golgi area. In COVNET45, the red markers represent the two objects X and Y, are shown by Fig 1B(II).

Fig 1B(III) shows the boxes labeled "S", which are states of those objects. Moreover, the boxes labeled "E" are the events in which those objects participate. The series numbers (1,2,3,

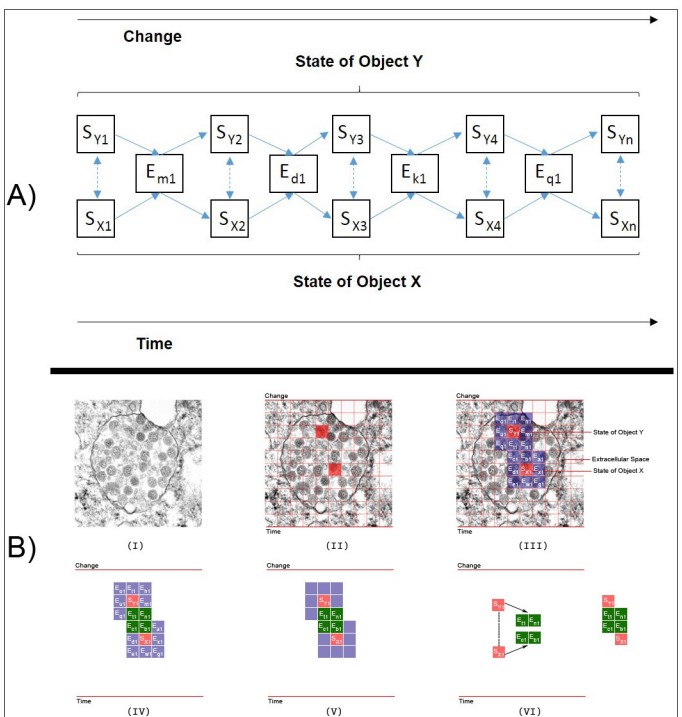

**Fig 1. Basic framework using a sample image thin-section electron microscopic image of SARS-CoV-2.** A) Represents the relational paradigm ontology of objects, events and change. Reprinted from [26] under a CC BY license, with permission from CDC/ Cynthia S Goldsmith, original copyright 2020: none—this image is in the public domain and thus free of any copyright restrictions. B) Demonstrates the framework in thin-section electron microscopic image of SARS-CoV-2.

n) in sets of boxes indicate sequences. The Y series in the top row of the state boxes indicate the states of object Y; similarly for the bottom row of object X and its states. The alphabetic on the event boxes indicate different types of events (marked in blue).

Each object can participate in different types of events (marked in red). The events of states Y and X that superseded with each other are the events of virus particles shedding (marked in green in Fig 1B(IV) and 1B(V). The solid arrow shows each state of an object being superseded by a new state as the result of the object participating in the event. The double dashed-line arrows indicate states of objects that can affect and be affected by other states of objects, as seen in Fig 1B(VI). Moreover, some objects can take part in events without being changed. For instance, an object in specific events can cause other objects to be changed.

The diagram of COVNET45 in Fig 1A shows that only current states of objects can affect and be affected by other states. Past states of objects do not affect anything, except through their effect, directly or indirectly, on current states. Future states of objects do not affect anything because future states do not yet exist, and what does not exist cannot affect anything (although, of course, the present anticipation of future states by sentient beings may affect current states of those sentient beings, and through them, the current states of other objects).

Change is what results from a process. It is what goes on "inside" an event. This process is a process of state transition. Before an object changes, it is in a given ephemeral state. It remains in that ephemeral state until it changes. After it changes, it is in a new ephemeral state, which may or may not also be a temporary state that it has been in before. These changes are what

goes on in the events that happen to the object. This process of state/change/new state occurs over and over again, from when the object begins to exist to when it no longer exists.

The next section explains the technique employed by COVNET45 in processing nearly any giving objects in 2D images such as microdroplets or virus particles. Later we show how COVNET45 can provide accurate measurements in different scale systems.

## Image utilization

COVNET45 can process different types of images specifically raster image files, since raster image files contain pixel values, and saved in an image file with JPG, JPEG, GIF, or PNG extension formats. COVNET45 also supports WEBP, JPS, JFIF, CUR, BMP, JPE and SVG image formats. In COVNET45 all uploaded images will remain unmodified and retain their original format; however, COVNET45 appends them internally to an existing Document Object Model (DOM) element for graphical representation reasons.

Fig 2 shows the extended framework version of COVNET45 for constructing and processing microscopic images into the tool. In the image construction phase, a referent is a type realized in any giving raster image at a time. Every referent (image) consists of pixels and each row of pixels consists of values. A referent is an object or an event. Therefore every pixel is an object or an event. And so the set represented by any pixel is either object of a specific type of events. In the mathematics phase, every bit of pixels is calculated in any giving image once uploaded to COVNET45. It calculates the length of the image in pixels and scales different types of bar values in pixels, and the resulting value is represented either as a value of object or an event or both.

Fig 3 shows a demonstration of an original microscopic image employed using the extended framework version of COVNET45 for constructing and processing raster images. The microscopic sample image demonstrates both states of objects Y and X (marked in red) with ten grid system blocks, and image size of 737-pixel in width and 737-pixel in height. The original image extension type is "TIF", and the original dimension pixel size is 1461-pixel in width by 1461-pixel in height. The referent is focusing only on the objects (two virus particles).

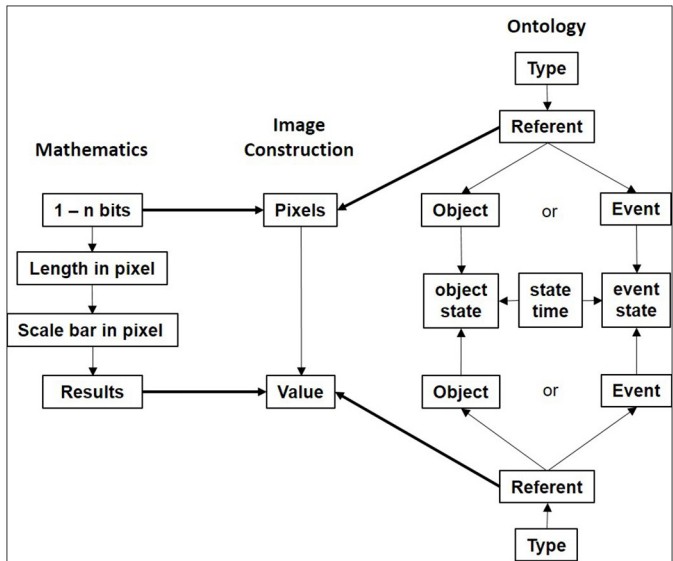

**Fig 2. Extended framework of COVNET45 for constructing and processing raster images.**

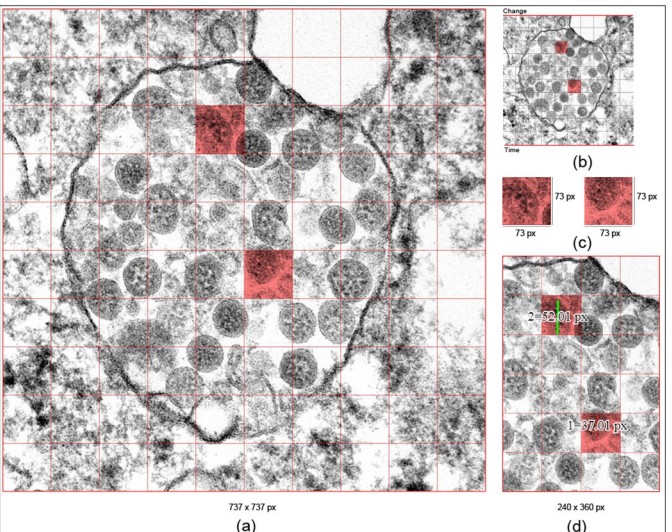

**Fig 3.** Sample microscopic image of COVID-19 virus, (a) Represent the pixel dimension of the 737x737 pixels image highlighted with states of objects X and Y, (b) Shows the referent process of the extended framework of processing only object states of X and Y, (c) Shows the pixel value of objects X and Y explaining the image construction phase, (d) Shows the mathematics phase of measuring only the dimension of the virus (objects) in states X and Y.

We use the ten grid system applied in the image to provide an understanding of COVNET45 frameworks. In image construction phase the grids highlighted in red (State of object Y and X) demonstrate the size of a single block in pixels, which also contains a single virus particle size, the dimensions of both states Y and X are 73-pixel width by 73-pixel height.

In COVNET45, we calculate the size of many objects' states; in this, we measure the size of a single virus particle in both states (Y and X). The size of the virus in object Y is 52.01-pixel in height. In contrast, the size of the virus in object X is the 37.01-pixel in height. Note that these measurements will be altered if an image has been resized or reshaped. In COVNET45, we use multi-layer processing points in each uploaded image file. A user can construct and process raster images independently for a single image at a time. Fig 4 explains the technique used in COVNET45 for generating a new image, and the processing is described in details below.

**Semantic processing.** The image preprocessing step is performed to process a single raster image uploaded to COVNET45 tool at a time. The left-hand side of the solution typically is the front end or channel referred to as (Image Type), this is strictly applied only to raster image format type of JPG, JPEG, GIF, PNG, WEBP, JPS, JFIF, CUR, BMP, JPE and SVG. The brain behind COVNET45 would be the multi-layered automation functionality services. At the core of COVNET45, it provides automation solutions, which calculates' measurements.

OpenCV includes is an open sources library with numerous pre-trained machine learning classifiers for computer vision problems such as object detection. OpenCV hides the complexity of machine learning such as feature selection, classifier selection and training, and other mathematical complexity. It is used as for detection in COVNET45.

Currently, COVNET45 supports five processing automation techniques described below.

- Image Segmentation (SEGBON): This automated feature uses an image segmentation technique adopting watershed algorithm, which provides an alteration on grayscale images to view it as a topographic surface. We present the watershed algorithm by using OpenCV programming module adopted as a marker-based watershed transformation; every image preprocessed using this algorithm will be executed as a topographic surface.

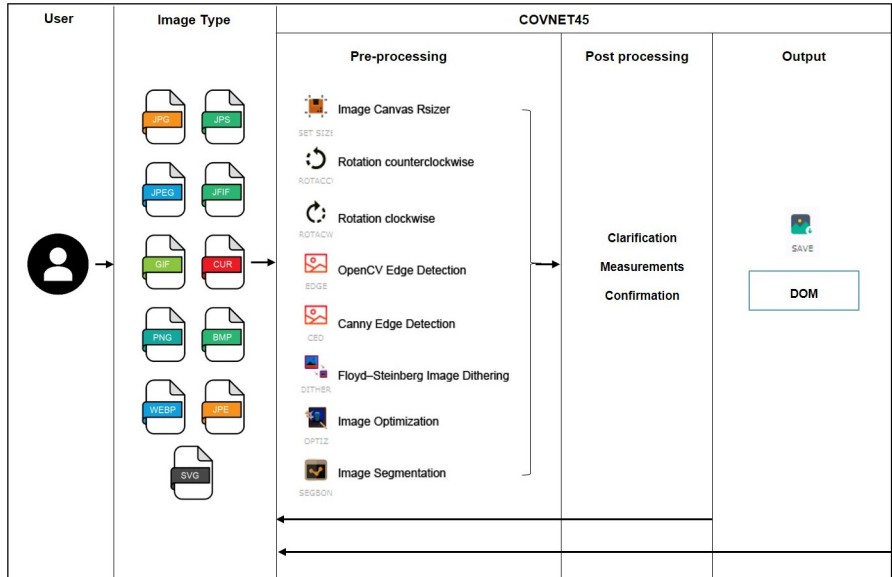

**Fig 4. COVNET45 methods for processing raster images.**

- OpenCV Edge Detection (EDGE): This automated feature uses John F. Canny, Canny edge detection algorithm, which is sensitive to noise that is eliminated using Gaussian filters to detect edges in the blurred image.

- Canny Edge Detection (CED): This automated feature uses a supervised version of OpenCV Edge Detection. An improvement to the previous technique, supervised CED introduces thresholds that can be added to a fine-tuned version of OpenCV Edge Detection, that are represented in sigma, thresholds, and Non-maximum suppression.

- Floyd-Steinberg Image Dithering (DITHER): This automated feature demonstrates an error-diffusion technique, intended to take advantage of binary images to increase the visual quality of the produced binary images. Where its calculations are based on quantization error between the pixel and its neighboring pixels when scanned neighbors of scanned pixels are divided by error diffusion filter weights. The tool is taking the Floyd, Steinberg approach by using input image preprocessing as an over quantization then dithered as a grayscale output as a PNG compressed output, user interaction is prompt into selecting images without any selection over algorithm weights.

- Image Optimization (OPTIZ): This automated feature similar to DITHER uses Floyd, Steinberg algorithm. However, the only difference is that it allows altering the images in grayscale output, as well as allowing scaling the algorithm weights.

**Post processing.** The image post-processing step is performed after completion of the preprocessing step. This step consists of three stages, which are clarification, measurements and confirmation. Each image uploaded to COVNET45 will be processed using a user selection type of automation features described above. Once the output is confirmed, the users are eligible to perform the calculation of any selected area measurements within the image pixels.

To maintain a precise measurement scale of two points or more in each object in every state, whether, in state of object Y (Sy1) or state of object X (Sx1) during the post-processing

stage, a right sampling approach over the states (X or Y) had to be implemented. The main idea is to perform a universal conversion calculation mechanism to determine the precise sampling of each object within two or more points in a single image. This will provide us with a significant verification of the size of objects in each image.

Definition 1 explains the measurements process. The definition represents the instances: G as green marker line, R as red marker line, M as calculated measurement per scale and S as the scale bar value. We assume that a single image consists of two markers to calculate a single object in a single uploaded image in COVNET45. Then COVNET45 generates a pixel value measurement for the red marker and green marker lines, which allows us to produce a measurements per scale. The resulting value of the measurements per scale when multiplied with the value of the integer scale bar value, will result in the accurate measurement of the object size within the same scale of the image used at time of upload to COVNET45.

$$\frac{G_L(px)}{R_L(px)} = M_{(per\ scale)} \times$$

$$S_{(scale\ bar)} = M_{(scale\ bar\ measurements)}$$

Fig 5(B) shows thin-section electron micrographs of the 2019 novel coronavirus grown in cells. We show the implementation of accurately measuring a single infected cell. Marked

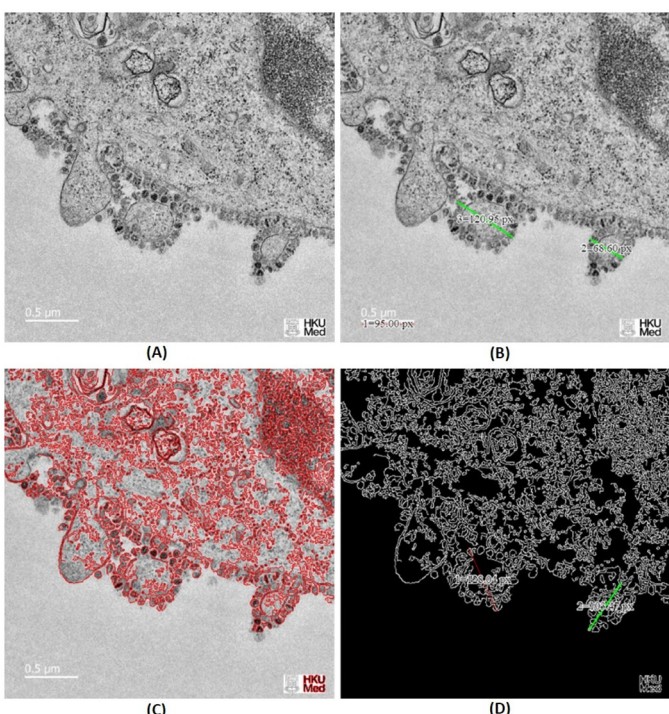

(A)  (B)

(C)  (D)

**Fig 5.** Demonstrates OpenCV Edge Detection feature in COVNET45: A) Original image shows thin-section electron micrographs of the 2019 novel coronavirus grown in cells published by the University of Hong Kong. The image shows part of a virus infected cell grown in culture with multiple virus particles being released from the cell surface. Each infected cell produces thousands of new infectious virus particles, which can go on to infect new cells. Reprinted from [27] under a CC BY license, with permission from LKS Faculty of Medicine, and Electron Microscopy Unit, The University of Hong Kong, original copyright 2019. B) Shows exported image marked with sample measurements in pixels to two infected cells with COVID-19 using COVNET45 tool. C) Shows the implementation of ridge detector to the original image in COVNET45 using the feature (EDGE). D) Shows the implementation of canny edge detector to the original image in COVNET45 with the use of non-maximum suppression technique using the feature (EDGE).

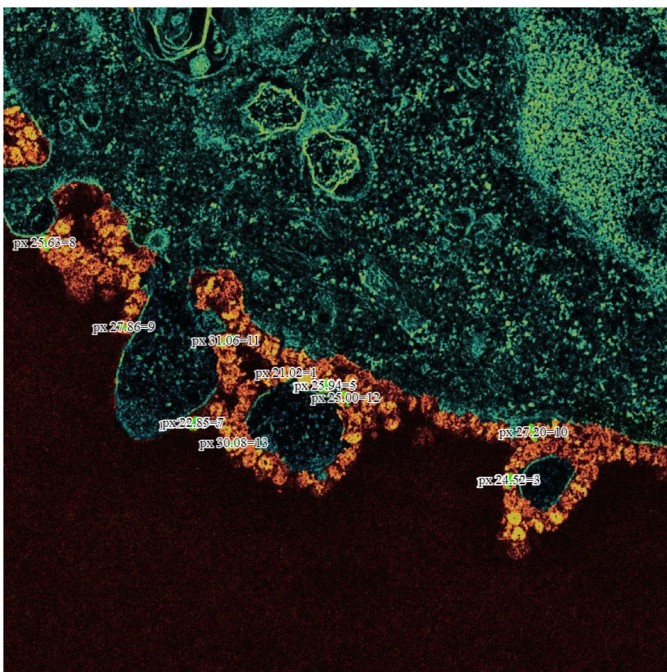

**Fig 6. Original image shows thin-section electron micrograph of the 2019 novel coronavirus grown in cells published by the University of Hong Kong.** The image shows part of a virus infected cell grown in culture with multiple virus particles being released from the cell surface. Reprinted from [27] under a CC BY license, with permission from LKS Faculty of Medicine, and Electron Microscopy Unit, The University of Hong Kong, original copyright 2019.

in green displacement diagonal line no. (2) and (3). After applying the formula, we can conclude the size of cell number (2) is 0.36 μm, while the size of cell number (3) is 0.63 μm. Moreover, In Fig 6, we demonstrate and validate the virus's particles estimated measurement, which is between 0.10 μm to 0.16 μm, which indicates that the virus particles size in an infected cell are approximated and can be measured correctly using electron micrographs microscope.

In terms of measurements in different scale systems, the formula allows the scale to match the final results with the same image scale used in the photograph. In Fig 7(C), we notice the image scale is on a centimeters scale. The research concludes that it corresponds to droplet size to nuclei of ca. 4 μm diameter, or 12- to 21-μm droplets before dehydration. Analyzing the highest droplet measurements in the image resulted in a different size than the research suggested. Marker sample number 2 considered the longest droplet in the image with a measurement size of 3.70 cm, and sample number 3 shows a tiny droplet with a measurement size of 0.23 cm. We conclude that COVNET45 uses the image scale, revealing the answer to the droplets' correct measurements.

## Implementation and experimental evaluation

COVNET45 uses standard web technologies and is hosted on AWS cloud servers. The software presented as Software-as-a-Service (SaaS) cloud-based software architecture, built with the latest Codeignitor V3 framework. Below we discuss the features of the experiments provided by COVNET45.

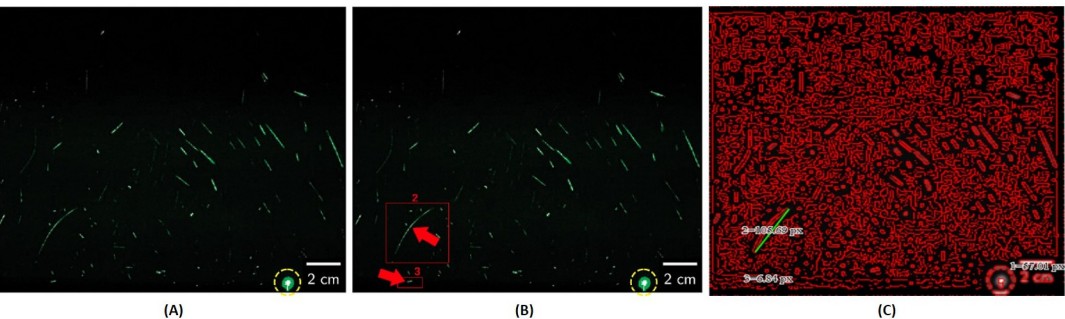

**Fig 7.** Demonstrates Canny Edge Detection feature of COVNET45: A) Original image shows speech droplets using highly sensitive laser light, the image describe the observation of airborne speech droplet nuclei, generated by a 25-s burst of repeatedly speaking the phrase "stay healthy" in a loud voice, the droplet size to nuclei is 4 μm diameter, or 12- to 21-μm droplets prior to dehydration. Reprinted from [3] under a CC BY license, with permission from Laboratory of Chemical Physics, National Institute of Diabetes and Digestive and Kidney Diseases, National Institutes of Health, Bethesda, MD 20892-0520, original copyright 2020. B) The marker demonstrates the test measurements that COVNET45 will verify. C) Shows iteration of the image using the CED technique in COVNET45, which reveals the edges of the droplets using high intensity thresholds. The real measurements of the sample number 2 considered the longest droplet in the image with size of 3.70 cm and sample number 3 shows a small droplet with measurement size of 0.23 cm.

## Experimental results

The following subsections used several real images to evaluate five different features of COVNET45.

**1. Image segmentation.** Fig 8 demonstrates the results of two test images validating the use of Image Segmentation approach in COVNET45. It shows the use of image segmentation approach over watershed transformation, which allows adding foreground extraction using the "GrabCut" algorithm. This allows regions outside the box boundaries of foreground, then segments the content within the box region. The user interaction selecting objects to assign them as a foreground or background [28].

Fig 8(A) shows the results of the foreground segmentation based on "GrabCut" nuclei images by eliminating the box region. Processing the original image in COVNET45 the result in Fig 8(B) shows that overlapped particles and spaces are not segmented by applying the mask, and boundaries have been judged based on the segmented objects' edges [29].

Results of processing test images in Fig 8(C) and 8(D) reveal similarity in regions of interest, since those regions are segmented similarly. This case can be generalized to similar cases in extracting segments of interest from similar medical images, where the iterative approach is segmenting the object by border matting around the rigid segmentation boundary [30].

**2. OpenCV Edge Detection.** Fig 5 Demonstrates the results of single test image validating the use of OpenCV Edge Detection (EDGE) algorithm approach in COVNET45. Fig 5(A) shows the original image of thin-section electron micrograph of the 2019 novel coronavirus grown in cells. The image demonstrates part of the virus-infected cell grown in a culture with multiple virus particles being released from the cell surface.

Moreover, Fig 5(C) and 5(D) demonstrate the technique used in COVNET45 using ridge detector and canny edge detector, revealing more information in the image edges than the original image. Detecting edges can help measure cells with higher precision compared to the original image.

**3. Canny Edge Detection.** Similarly, Fig 7 demonstrates the feature Canny Edge Detection, which uses the same technique as the OpenCV Edge Detection. However, the regions in any uploaded image are more automated in CED depending on the threshold provided when

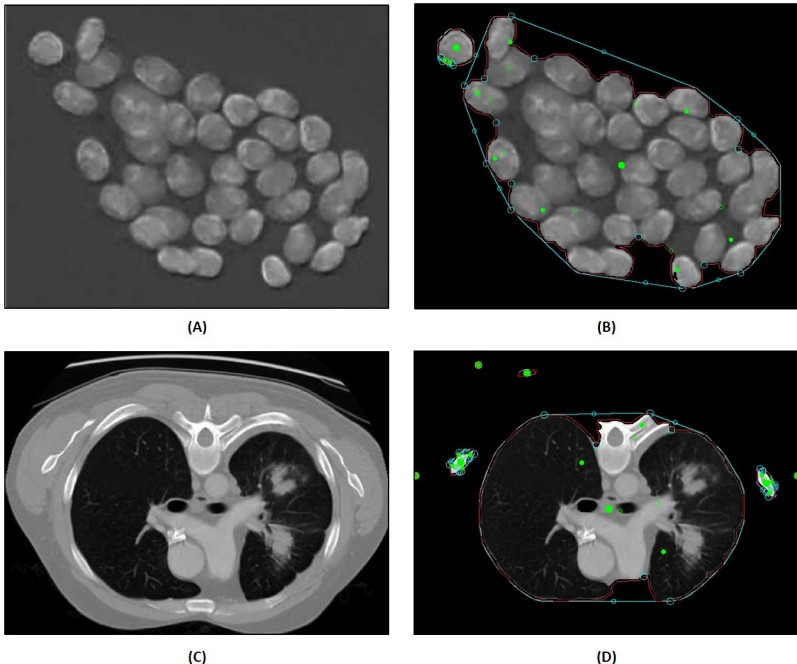

**Fig 8.** Demonstrates image segmentation feature in COVNET45: A) The original image of hematoxylin-stained objects (nuclei). Reprinted from [29] under a CC BY license, with permission from Institute of Control and Computation Engineering, University of Zielona Góra, Zielona Góra, Poland, original copyright 2020. B) Shows the output image after applying segbon segmentation in COVNET45. C) Demonstrates the original input image of the vertical lung scan of automatic lung tumor segmentation on PET/CT images using fuzzy Markov random field model. Reprinted from [30] under a CC BY license, with permission from Tianjin Key Lab of BME Measurement, Tianjin University, Tianjin 300072, China, original copyright 2014. D) Shows the output image after applying SEGON segmentation in COVNET45.

uploading the image. Canny Edge Detection can refine extra details in the images by the user based thresholds, which help reveal the darker shades of dark regions in the image. The marked regions 2 and 3 are examples of applying thresholds to reveal the cells' details, so the measure tool can be applied to measure the length of cells.

**4. Floyd-Steinberg image dithering.** Fig 9 demonstrates the Floyd-Steinberg Image Dithering feature of COVNET45. Fig 9(A) shows the original image recording of a real person coughing as a test subject. It demonstrates that air droplets can travel 1 m away from the mouth [17]. Fig 9(B) demonstrates the original image using the Floyd Steinberg algorithm distortion to generate additional details on the image. This technique is used in COVNET45 to add extra sharpness to the image, which reveals more details of the air droplet particles. Added sharpness in dithering reduces the error artifacts when low-quality images are inputted to the tool, allowing the recovery of the information lost to under-sampling by pixel in images that can be used to recover the details of imported images into COVNET45.

**5. Image optimization.** Fig 10 demonstrates the Image Optimization feature of COV-NET45. Fig 10(A) shows representative fluorescent micrograph of fibrous or cellular deposits in the plasma smears of COVID-19 patients [18]. This feature result is similar to error diffusion using DITHER algorithm. This feature allows the users to customize the image from background colors and delete certain features from the uploaded image. Uploaded images into this feature provide transparency to the background, which highlights the objects in the images. Exported images using OPTIZ can be stored at either black and white, inverted black and white in 2, 4 or 8 bits per pixel (bpp).

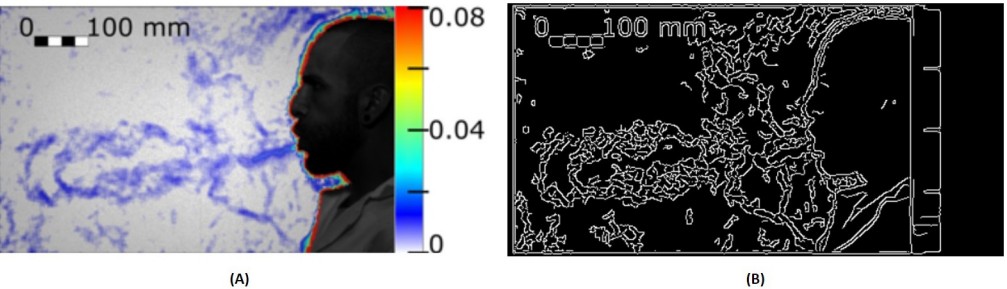

**Fig 9.** Demonstrates Floyd-Steinberg Image Dithering feature in COVNET45: A) Original image shows the test of a real person coughing, which generates an airflow. The image was taken using a high-speed CMOS camera (VEO710L, Vision Research) with 1280 x 800 square pixel, recording the displacement of the droplets is measured as 1 meter away from the mouth. Reprinted from [17] under a CC BY license, original copyright 2020. B) Shows exported image using error diffusion using the Floyd Steinberg dithering algorithm using COVNET45 tool, revealing more characteristics of air droplets particles in the image.

## Experimental setup and implementation

In order to test the tool, we used several images from previously published research as seen in Figs 3 and 5–10. The experiments and development were carried on Fujitsu Siemens Lifebook laptop model AH532/G21, with 8GB of memory, 2.6 GHz Intel Core i5 processor with HD graphics 4000 Nvidia Geforce. The web application is built as SaaS hosted in AWS cloud with AES 256 encryption for added privacy. COVNET45 tool was tested using Google Chrome version 88 and Firefox version 72. COVNET45 is coded using the Application Development Framework (CodeIgnitor V.4) using PHP and MySQL 5.1+ as a database structure.

COVNET45 synthesis is a robust solution to optimize and measure a variety of medical and non-medical raster images types. Fig 11 shows COVNET45 dashboard and application GUI screens in responsive and web views. COVNET45 was able to authenticate and measure high accuracy measurement results by optimizing raster image types. COVNET45 combines strong capabilities of optimizing raster images with automation detection mechanisms and accurately measuring different scale systems. Furthermore, COVNET45 handles only a single image as

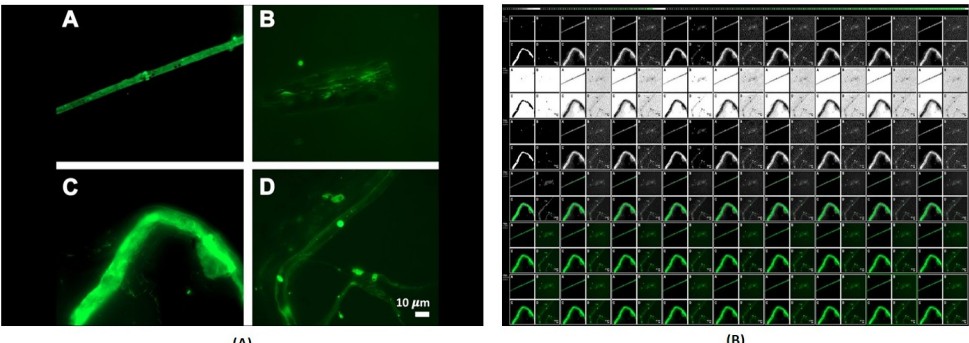

**Fig 10.** Demonstrates Image Optimization feature of COVNET45: A) Original image shows representative fluorescent micrographs of fibrous or cellular deposits in the plasma smears of COVID-19 patients. The image was taken from the research prevalence of amyloid blood clots in COVID-19 plasma. Reprinted from [18] under a CC BY license, with permission from Department of Physiological Sciences, Faculty of Science, Stellenbosch University, Stellenbosch, Private Bag X1 Matieland, 7602, South Africa, original copyright 2020. B) Shows the exported images using OPTIZ in COVNET45 tool.

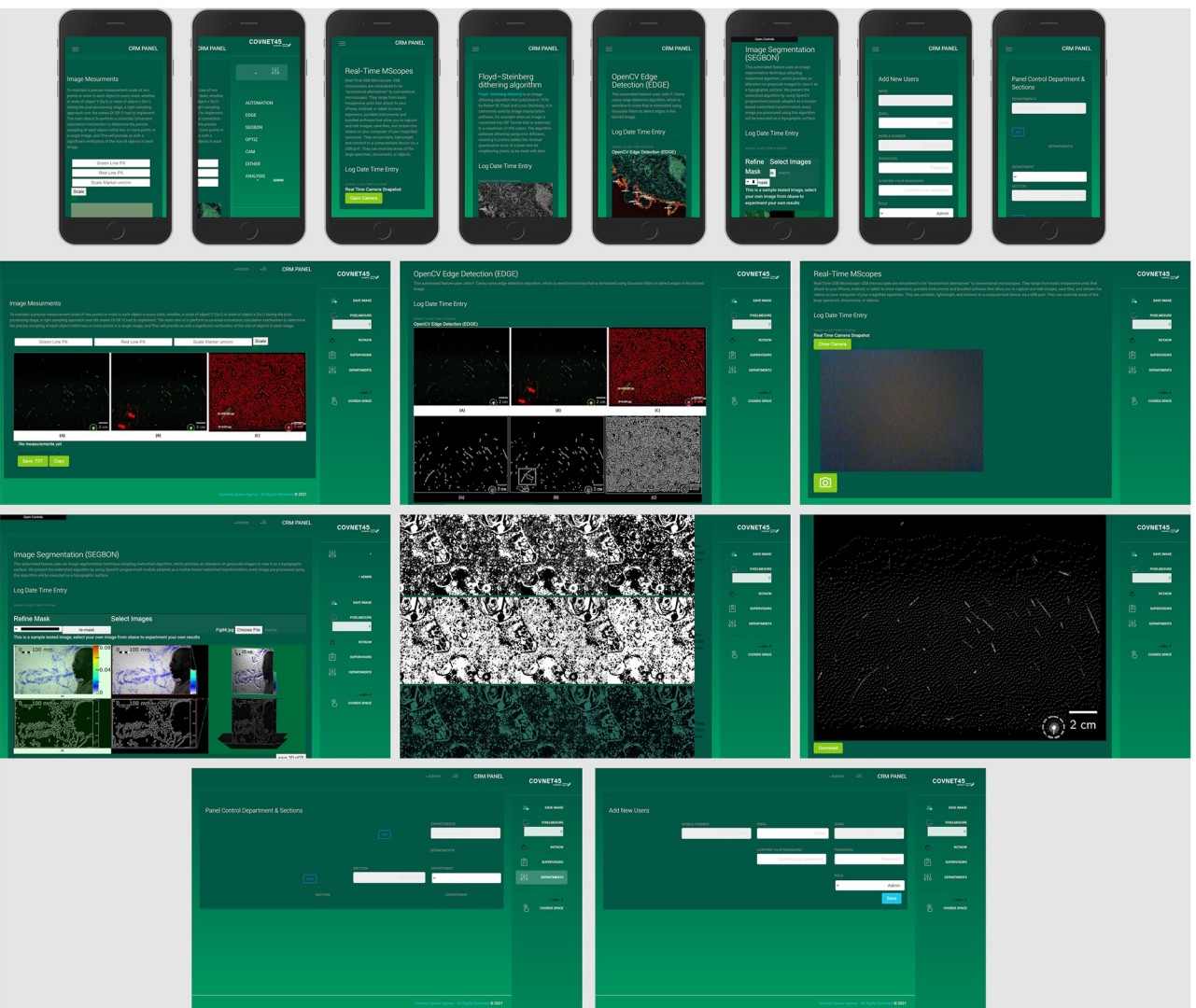

**Fig 11. COVNET45 dashboard and application screens in responsive and web views.** The screens demonstrate the machine learning algorithms screens and the control panel for administration access privileges.

an input terminal upload and "is limited to the raster image types of JPG, JPEG, GIF, and PNG formats. Finally, COVNET45 supports WEBP, JPS, JFIF, CUR, BMP, JPE and SVG image formats.

## Limitations

COVNET45 has the following limitations.

- Image segmentation feature is unable to process certain objects if the image's foreground is not completely black, therefore the algorithm won't distinguish small particles that contain a similar color to the image foreground.

- COVNET45 uses different machine learning algorithms in each feature, which may slow the automation done in the cloud and consequently the processing in the client DOM browser.

- COVNET45 uses image scale bars to measure accurately any given object in the image; without the scale system applied in the image, COVNET45 won't be able to distinguish the level scale system to be used.

Moreover, respiratory transmission depends on the incorporation of the airborne particles in aerosols. Aerosols are produced during speaking and regular breathing, while coughing produces even more forceful expulsion. Transmission from the nasal cavity is facilitated by sneezing and is much more effective if the infection induces nasal secretions. A sneeze produces up to 20,000 droplets (in contrast to several hundred expelled by coughing), and all may contain rhinovirus if the individual has a common cold. As noted when we discussed viral entry, the size of a droplet affects its "hang time": large droplets fall to the ground, but smaller droplets (1 to 4 um in diameter) may remain suspended in the air for a longer time. Nasal secretions also frequently contaminate hands or tissues. The infection may be transmitted when these objects contact another person's fingers, and that person, in turn, touches his or her nose or conjunctiva. In today's crowded cities, transport and workplaces, people's physical proximity may facilitate viruses to spread more effectively. In Fig 1 we show how COVNET45 would be able to detect events of particles spread in cells, where a single spherical virus particle contains black dots of COVID-19 virus as an *object* in a single microscopic image, and the possible movements/diameter in size of a single spherical virus particle as *events*. Spherical virus particles contain black dots, which are cross-sections through the viral nucleocapsid. In the cytoplasm of the infected cell, clusters of particles are found within the membrane-bound cisternae of the rough endoplasmic Reticulum/Golgi area. This will allow us to develop a pattern recognition model for the behavior and spread in micrograph images, and measure cells infection before the virus spread into the human body.

## Conclusions and future work

Biologists use microscopic images to study biological data to analyze cellular structure and organism characteristics. Artificial Intelligence algorithmic may assist in understanding different cell characteristic under different conditions such as germ, viruses and effect of cell proliferation.

This paper presented an automatic measurement detection tool, dubbed "COVNET45". The tool provides five automatic intelligent techniques (SEGBON, EDGE, CED, DITHER and OPTIZ) to process, detect and measure any given objects and events in raster image files. COVNET45 will allow scientists to automatically optimize raster images, such as micrograph images of microscopic images, and perform precise measurements observation to any given object or particles. In addition, COVNET45 can examine anatomical cell and measure characteristics like capturing and tracking cells in microscopic studies.

The experimental results demonstrated that COVNET45 is helpful in processing high-resolution images and exploring detailed or hidden characteristics. We demonstrated accurate measurements of virus particles in thin-section electron micrographs using the EDGE feature. In addition, COVNET45 allows measurements in different scales, enabling the scales to match the final results with the same image scale. It also uses error diffusion using the Floyd Steinberg algorithm revealing more characteristics of the air droplets particles in the image. The Floyd Steinberg dithering add extra sharpness and reduces error in low-quality images allowing to recover lost data due to sampling.

The experiments also demonstrated the use of image segmentation approach over watershed transformation allowing foreground extraction using the GrabCut algorithm, which showed the clear boundaries given that the particles and spaces are not segmented. Moreover the use of OpenCV EDGE algorithm reveals more information in the COVNET45 image

edges than the original image, which can help measure cells with higher precision. Experiments showed that CED can refine extra details in the images, which reveals the darker shades helping in revealing the cell details and measuring its length. Finally the OPTIZ image optimization feature of COVNET45 allows users to customize the images and delete certain features. This adds transparency to the background highlighting the objects in the image.

COVNET45 is a work in progress to allow users to upload multiple images as opposed to a single upload in the current version. Additionally, we plan to add more add more graphical features to control images, provide markup comments and highlight selection from the images. We are working on developing an internal mechanism to minimize external calls to third-party modules. COVNET45 is currently being integrated with DOM scripting, eliminating uploads and avoiding storage of images in the server, which provides efficiency in image optimization automation. Finally, COVNET45 is currently hosted in AWS and can be integrated with the Amazon artificial intelligence machine learning platform. This will allow the tool to transmit and perform optimization to the images more effectively and enhance security.

## Author Contributions

**Conceptualization:** Monther Aldwairi, Sandra C. P. Cachinho.

**Data curation:** Faten Yasin.

**Formal analysis:** Faten Yasin, Sandra C. P. Cachinho.

**Funding acquisition:** Monther Aldwairi.

**Investigation:** Sandra C. P. Cachinho, Abdullah Hussein.

**Methodology:** Hesham H. Alsaadi, Monther Aldwairi, Faten Yasin, Abdullah Hussein.

**Project administration:** Abdullah Hussein.

**Resources:** Monther Aldwairi, Sandra C. P. Cachinho, Abdullah Hussein.

**Software:** Hesham H. Alsaadi, Sandra C. P. Cachinho.

**Supervision:** Monther Aldwairi, Sandra C. P. Cachinho.

**Validation:** Hesham H. Alsaadi, Faten Yasin, Abdullah Hussein.

**Visualization:** Hesham H. Alsaadi.

**Writing – original draft:** Hesham H. Alsaadi, Faten Yasin, Sandra C. P. Cachinho.

**Writing – review & editing:** Monther Aldwairi.

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
