## [Decision Letter · Decision Letter 0]

2 Aug 2022

PONE-D-22-15591Artificial intelligence tool for the study of COVID-19 microdroplet spread across the human diameter and airborne spacePLOS ONE

Dear Dr. Aldwairi,

Thank you for submitting your manuscript to PLOS ONE. After careful consideration, we feel that it has merit but does not fully meet PLOS ONE’s publication criteria as it currently stands. Therefore, we invite you to submit a revised version of the manuscript that addresses the points raised during the review process.

We look forward to receiving your revised manuscript.

Kind regards,

Anandakumar Haldorai, PhD

Academic Editor

PLOS ONE

Journal Requirements:

MA

Research Incentive Fund Grant #R20089

Zayed University Research Office

www.zu.ac.ae

NO.  

This work was supported by Zayed University Research Office, Research Incentive Fund

Grant #R20089

 MA

Research Incentive Fund Grant #R20089

Zayed University Research Office

www.zu.ac.ae

NO

5. Please upload a copy of Figure ??, to which you refer in your text on pages 5 and 12. If the figure is no longer to be included as part of the submission please remove all reference to it within the text.

6. Please ensure that you refer to Figure 11 in your text as, if accepted, production will need this reference to link the reader to the figure.

7. We note that Figures 2, 4, 5, 6, 7, 8 and 9 in your submission contain copyrighted images. All PLOS content is published under the Creative Commons Attribution License (CC BY 4.0), which means that the manuscript, images, and Supporting Information files will be freely available online, and any third party is permitted to access, download, copy, distribute, and use these materials in any way, even commercially, with proper attribution. For more information, see our copyright guidelines: http://journals.plos.org/plosone/s/licenses-and-copyright.

a) You may seek permission from the original copyright holder of Figures 2, 4, 5, 6, 7, 8 and 9 to publish the content specifically under the CC BY 4.0 license. 

Additional Editor Comments:

There is no new innovations with proposed framework, Whatever stated is commonly available with all the COVID-19 microdroplet spread.

How the super speech emitters having specimens and feature fusion model proposed with classifiers.

The quality of language satisfies. But the manuscript needs to be edited for grammar and syntax.

Spell out each acronym the first time used in the body of the paper. Spell out acronyms in the Abstract only if used there.

Make sure that the Conclusion briefly summarizes the results of the paper it should not repeat phrases from the Introduction.

Reviewers' comments:

Reviewer's Responses to Questions

**Comments to the Author**

1. Is the manuscript technically sound, and do the data support the conclusions?

Reviewer #1: Yes

Reviewer #2: Partly

2. Has the statistical analysis been performed appropriately and rigorously? 

Reviewer #1: N/A

Reviewer #2: No

3. Have the authors made all data underlying the findings in their manuscript fully available?

Reviewer #1: Yes

Reviewer #2: Yes

4. Is the manuscript presented in an intelligible fashion and written in standard English?

Reviewer #1: No

Reviewer #2: Yes

5. Review Comments to the Author

Reviewer #1: The authors proposal for an artificial intelligence base detection of potential viral spread following the sneezing, coughing, and talking appears very interesting. However, the authors are required to make major revisions to the manuscript in terms of clarity of presentation. In the present form, the methodology of the COVNET45, its performance, and the interpretation of the results appear unclear. Also, it is not clear how this would benefit the scientific community and the governments to control the COVID-19 pandemic. Moreover, does this application has benefits in other diseases, etc. is not explored by the authors.

An in-depth and professional language editing may be essential

Reviewer #2: We understand that the COVID-19 pandemic has made an impact on infectious disease testing and the need to expand testing across the continuum of care. Creation of new tools for virus testing is fully reasonable.

This article devoted of using the specific software for images and drops measurement which appear in the process sneezing or cough or breathing or talking. These drops might appear not only in COVID-19 infected persons, but in influenza and other respiratory diseases infected persons (not only viral but also bacterial infections). Therefore - the title of the article with COVID-19 not appropriate. Perhaps this new program software may be use for different aims, but it not allowed to differentiation of drops of one infectious diseases from other. My proposition for authors - to corrected this article.

Besides in the article on the pages 5 and 12 not indicated the Figure's number (Fig.??).

6. PLOS authors have the option to publish the peer review history of their article (what does this mean?). If published, this will include your full peer review and any attached files.

Reviewer #1: **Yes: **Dr. Venkataramana Kandi

Reviewer #2: No

---

## [Author Response · Author response to Decision Letter 0]

4 Oct 2022

Summary of Revision

Dear reviewers,

 We deeply appreciate your valuable and constructive comments and we thank you for the review, because it ultimately helped improve the manuscript. We worked diligently and made significant efforts to address and to carry out all essential revisions suggested by you. We are confident that this revised manuscript reflects most of the reviewers’ comments. 

We hope this revised version is satisfactory and will lead to a positive decision. We detail our responses below.

Editorial Comments

We thank the editor for his valuable comments.

We are using PLOS ONE LaTeX templates and adhering to the style requirements referenced above.

Action and changes to the paper

Authors emails have been removed from authors list and formatting fixed

MA

Research Incentive Fund Grant #R20089

Zayed University Research Office

www.zu.ac.ae

NO. 

The funders did not play any role and we added the statement to the cover letter.

Please update the submission from your side.

This work was supported by Zayed University Research Office, Research Incentive Fund

Grant #R20089

 MA

Research Incentive Fund Grant #R20089

Zayed University Research Office

www.zu.ac.ae

NO

This is the only funding information, we removed it from acknowledgement section.

Action and changes to the paper

Removed from acknowledgement

I believe what we meant was the code for COVNET45 tools. All the data used is in the public domain!

Please change our Data Availability statement accordingly.

5. Please upload a copy of Figure ??, to which you refer in your text on pages 5 and 12. If the figure is no longer to be included as part of the submission please remove all reference to it within the text.

The reference to Figure 1 was fixed, and consequently the references to the figure in pages 5 and 12 were fixed.

Action and changes to the paper

Uncommented the Figure in LaTeX and commended the figure upload

6. Please ensure that you refer to Figure 11 in your text as, if accepted, production will need this reference to link the reader to the figure.

When we fixed the reference to Figure 1, all Figure numbers have been changed.

Figure 11 is now referenced on page 11.

7. We note that Figures 2, 4, 5, 6, 7, 8 and 9 in your submission contain copyrighted images. All PLOS content is published under the Creative Commons Attribution License (CC BY 4.0), which means that the manuscript, images, and Supporting Information files will be freely available online, and any third party is permitted to access, download, copy, distribute, and use these materials in any way, even commercially, with proper attribution. For more information, see our copyright guidelines: http://journals.plos.org/plosone/s/licenses-and-copyright. 

Please note that all figure numbers have been changed according to the citations.

All of the figures are available under open license as follows (Except one figure that we obtained the permission from the owner) (used in Figs 5 and 6).

Fig1.(B-I) and Fig3.(a) 

IMAGE LINK: https://phil.cdc.gov//PHIL_Images/23591/23591_lores.jpg

IMAGE WEBSITE: https://phil.cdc.gov/Details.aspx?pid=23591

Copyright: Attribution 4.0 International (CC BY 4.0) 

Pls check the owner clarification Attached and I quote “Thank you for contacting us with your request to use material published in Preventing Chronic Disease (PCD). PCD is a government publication in the public domain. No copyright exists for published material and all content may be used, shared, and distributed as you wish; provided that proper credit is given to the author(s) of the article(s) and Preventing Chronic Disease. We recommend using the suggested citation provided in the article itself. Please let us know if you have any further questions or concerns”

Fig.2 is ours.

Fig.4 is ours.

Fig.5 (A) (previously Fig.6)

IMAGE LINK: https://covid19.med.hku.hk/-/media/K2/Downloadables/The-Virus/nCoV2019-Square.ashx?la=en

IMAGE WEBSITE: https://covid19.med.hku.hk/en/downloads/virus

Copyright: We obtained the permission of the owner (JOHN NICHOLLS). (email attached) 

Figs 6 (previously Fig 10)

Copyright: We obtained the permission of the owner (JOHN NICHOLLS). (email attached) 

Fig.7 (A) (previously Fig.8)

Viola, I. M., Peterson, B., Pisetta, G., Pavar, G., Akhtar, H., Menoloascina, F., ... & Mehendale, F. V. (2021). Face coverings, aerosol dispersion and mitigation of virus transmission risk. IEEE Open Journal of Engineering in Medicine and Biology, 2, 26-35. 

PAPER: https://ieeexplore.ieee.org/abstract/document/9329130

IMAGE: https://ieeexplore.ieee.org/mediastore_new/IEEE/content/media/8782705/9299466/9329130/viola3-3053215-large.gif

COPYRIGHT: Attribution 4.0 International (CC BY 4.0) 

Fig.8 (A) (previously Fig 5)

Kowal, M., Żejmo, M., Skobel, M., Korbicz, J., & Monczak, R. (2020). Cell nuclei segmentation in cytological images using convolutional neural network and seeded watershed algorithm. Journal of Digital Imaging, 33(1), 231-242. 

PAPER: https://link.springer.com/article/10.1007/s10278-019-00200-8

FIGURE LINK: https://link.springer.com/article/10.1007/s10278-019-00200-8/figures/7

Copyright: Attribution 4.0 International (CC BY 4.0) 

Fig.8 (C) (previously Fig 5)

Cai, Y., Huang, T., Chen, L., Gao, S., & Zhang, N. (2014). Novel computational methods and tools in biomedicine and biopharmacy. Computational and Mathematical Methods in Medicine, 2014. 

PAPER: https://www.hindawi.com/journals/cmmm/2014/401201/

FIGURE LINK: https://www.hindawi.com/journals/cmmm/2014/401201/fig1/

copyright: Copyright © 2014 Yu Guo et al. This is an open access article distributed under the Creative Commons Attribution License, which permits unrestricted use, distribution, and reproduction in any medium, provided the original work is properly cited. 

Fig.9 (A,B) (previously Fig.7)

Stadnytskyi, V., Bax, C. E., Bax, A., & Anfinrud, P. (2020). The airborne lifetime of small speech droplets and their potential importance in SARS-CoV-2 transmission. Proceedings of the National Academy of Sciences, 117(22), 11875-11877. 

PAPER: https://www.pnas.org/doi/full/10.1073/pnas.2006874117

FIGURE LINK: https://www.pnas.org/cms/10.1073/pnas.2006874117/asset/03a58387-760c-4c37-9d6a-bb2fef5bc447/assets/graphic/pnas.2006874117fig01.jpeg

Copyright: Copyright © 2020 the Author(s). Published by PNAS. This open access article is distributed under Creative Commons Attribution License 4.0 (CC BY). 

Fig.10 (A,B,C,D) (previously Fig.9)

Pretorius, E., Venter, C., Laubscher, G. J., Lourens, P. J., Steenkamp, J., & Kell, D. B. (2020). Prevalence of amyloid blood clots in COVID-19 plasma. 

PAPER: https://www.medrxiv.org/content/10.1101/2020.07.28.20163543v1.full-text

FIGURE LINK: https://www.medrxiv.org/content/medrxiv/early/2020/07/29/2020.07.28.20163543/F4.large.jpg

COPYRIGHT: Attribution 4.0 International (CC BY 4.0)

Action and changes to the paper

The figure captions of the copyrighted figures was modified to show the following text.

Reprinted from [20] under a CC BY license, with permission from LKS Faculty of Medicine, and Electron Microscopy Unit, The University of Hong Kong, original copyright 2019.

Additional Editor Comments:

1. There is no new innovations with proposed framework, Whatever stated is commonly available with all the COVID-19 microdroplet spread.

The article proposes and evaluates a complete framework and software that uses five techniques as well as machine learning for image optimization, object detection, visualization and measurement. This definitely sets it apart from the other image analysis work in the literature as discussed in the related work section. 

The main aim is to develop a novel image optimization technique for measuring the distance of microdroplets spread and transmissions in airborne space.

Action and changes to the paper

The contributions were modified to highlight the unique features of this framework as well highlighting the use of machine learning as follows. Pls check the response to reviewers 1 below for more details.

OpenCV includes is an open sources library with numerous pre-trained machine learning classifiers for computer vision problems such as object detection. OpenCV hides the complexity of machine learning such as feature selection, classifier selection and training, and other mathematical complexity. It is used as for detection in COVNET45.

2. How the super speech emitters having specimens and feature fusion model proposed with classifiers.

Our work is not concerned directly with advancing Covid-19 and transmission mechanisms from the medical and biological perspectives. However, we are focused on images processing models that may assist and enable physicians and other medical partitioners diagnose Covid-19 and analyze the spread patterns.

3. The quality of language satisfies. But the manuscript needs to be edited for grammar and syntax.

Noted.

Action and changes to the paper

The manuscript has been thoroughly proofread (see marked version)

4. Spell out each acronym the first time used in the body of the paper. Spell out acronyms in the Abstract only if used there.

Noted.

Action and changes to the paper

Revised and defined all acronyms.

5. Make sure that the Conclusion briefly summarizes the results of the paper it should not repeat phrases from the Introduction.

Done.

Action and changes to the paper

Conclusions section has been revised.

Reviewer #1: 

1. The authors are required to make major revisions to the manuscript in terms of clarity of presentation. In the present form, the methodology of the COVNET45, its performance, and the interpretation of the results appear unclear. 

We proofread the manuscript and edited many paragraphs from the methodology, performance and results interpretation.

We feel that the technical description of the framework, while challenging to none computer scientists to follow, it is still clear to those with expertise in the field of raster image processing.

Action

The manuscript has been thoroughly proofread.

2. Also, it is not clear how this would benefit the scientific community and the governments to control the COVID-19 pandemic. Moreover, does this application has benefits in other diseases, etc. is not explored by the authors.

An in-depth and professional language editing may be essential.

COVNET45 explores the design aspect of a framework to be applied in the research in medical practices, the selection of the specific algorithms that has been proposed such as SOTA in their relative fields have been embedded in the design aspect of the COVNET45 application framework. We proposed and evaluated a complete framework and software that uses five techniques as well as machine learning for image optimization, object detection, visualization and measurement. COVNET45 used OpenCV with numerous pre-trained machine learning classifiers to perform object detection. 

It is true that the technique and tool may be useful in other airborne diseases. However, the selection of COVID-19 was due to the interest of authors to design a framework around it. The tested cases of Covid are selected to be positive in this research to validate the data and can be assisted later to design a different approach of a framework that can be tested for classification. Finally, other diseases can be tested in future.

Action and changes to the paper

The contributions have been rewritten as follows.

In this paper, we propose an intelligent technique to visualize, detect, and measure the distance of the spread of microdroplet transmissions in airborne space in a real-world setting. The article proposes and evaluates a complete framework and software that uses five techniques as well as machine learning for image optimization, object detection, visualization and measurement. The main aim is to develop a novel image optimization technique for measuring the distance of microdroplets spread and transmissions in airborne space.

The main contributions of this work are:

1. Generate high accuracy measurement results by optimizing the collected images and their characterization.

2. Combine the strong capabilities of accurately measuring spread with the use of different scale systems: nominal, ordinal, interval and ratio.

3. Provide a generic approach that is not limited to small scale microdroplets transmissions, but also able to measure the diameter of spread in any space using 2D graphical images with the capabilities of analyzing, optimizing and detecting various semantic marks.

4. Provide a versatile tool capable of analyzing different images formats, which may enable researcher to authenticate previous research results and compare to related work

5. Latest AES-256 encryption is used to protect the hosted images in AWS cloud to ensure added privacy.

Reviewer #2: 

1. This article devoted of using the specific software for images and drops measurement which appear in the process sneezing or cough or breathing or talking. These drops might appear not only in COVID-19 infected persons, but in influenza and other respiratory diseases infected persons (not only viral but also bacterial infections). Therefore - the title of the article with COVID-19 not appropriate. Perhaps this new program software may be use for different aims, but it not allowed to differentiation of drops of one infectious diseases from other. My proposition for authors - to corrected this article.

We thank the reviewer for this observation. It is true that the technique and tool are useful for other airborne diseases, however it is the authors position is that the title is appropriate since the images used to study the efficiency of the technique was Covid-19 related.

The authors will continue to study the proposed technique and improve the tool through testing with other airborne diseases’ images.

Action and changes to the paper

N/A

2. Besides in the article on the pages 5 and 12 not indicated the Figure's number (Fig.??).

Action 

All figures’ references have been fixed.

---

## [Decision Letter · Decision Letter 1]

17 Jan 2023

PONE-D-22-15591R1Artificial intelligence tool for the study of COVID-19 microdroplet spread across the human diameter and airborne spacePLOS ONE

Dear Dr. Aldwairi,

Thank you for submitting your manuscript to PLOS ONE. After careful consideration, we feel that it has merit but does not fully meet PLOS ONE’s publication criteria as it currently stands. Therefore, we invite you to submit a revised version of the manuscript that addresses the points raised during the review process.

We look forward to receiving your revised manuscript.

Kind regards,

Anandakumar Haldorai, PhD

Academic Editor

PLOS ONE

Journal Requirements:

Reviewers' comments:

Reviewer's Responses to Questions

**Comments to the Author**

1. If the authors have adequately addressed your comments raised in a previous round of review and you feel that this manuscript is now acceptable for publication, you may indicate that here to bypass the “Comments to the Author” section, enter your conflict of interest statement in the “Confidential to Editor” section, and submit your "Accept" recommendation.

Reviewer #1: All comments have been addressed

Reviewer #3: (No Response)

2. Is the manuscript technically sound, and do the data support the conclusions?

Reviewer #1: Yes

Reviewer #3: (No Response)

3. Has the statistical analysis been performed appropriately and rigorously? 

Reviewer #1: N/A

Reviewer #3: (No Response)

4. Have the authors made all data underlying the findings in their manuscript fully available?

Reviewer #1: Yes

Reviewer #3: (No Response)

5. Is the manuscript presented in an intelligible fashion and written in standard English?

Reviewer #1: Yes

Reviewer #3: (No Response)

6. Review Comments to the Author

Reviewer #1: Thanks for making the necessary changes. Unfortunately, the non computer field scientists would still feel difficult to understand this paper

Reviewer #3: -The paper should be interesting ;;;

-it is a good idea to add more photos of measurements, sensors + arrows/labels what is what (if any);;;

-What is the result of the analysis?;;

-figures should have high quality. ;;;;;

-text should be formatted;;;;

-what will society have from the paper?;;

-labels of figures should be bigger;;;;

-please add photos of the application of the proposed research, 2-3 photos ;;;

-Is there a possibility to use the proposed research for other topics:

"Role of Hybrid Deep Neural Networks (HDNNs), Computed Tomography, and Chest X-rays for the Detection of COVID-19",;;;

"A Novel Method for COVID-19 Diagnosis Using Artificial Intelligence in Chest X-ray Images",;;;

-references should be from the web of science 2020-2022 (50% of all references, 30 references at least);;;

-please compare advantages/disadvantages of other approaches;;;

-Conclusion: point out what have you done;;;;

7. PLOS authors have the option to publish the peer review history of their article (what does this mean?). If published, this will include your full peer review and any attached files.

Reviewer #1: **Yes: **Dr. Venkataramana Kandi

Reviewer #3: No

---

## [Author Response · Author response to Decision Letter 1]

7 Feb 2023

Summary of Revision

Dear reviewers,

 We deeply appreciate your valuable and constructive comments and we thank you for the review, because it ultimately helped improve the manuscript. We worked diligently and made significant efforts to address and to carry out all essential revisions suggested by you. We are confident that this revised manuscript reflects most of the reviewers’ comments. 

We hope this revised version is satisfactory and will lead to a positive decision. We detail our responses below.

Reviewer #1: 

1. All comments have been addressed

Thank you.

2. Thanks for making the necessary changes. Unfortunately, the non-computer field scientists would still feel difficult to understand this paper

Action

This is a technical paper with the expected audience made of technology scientists working on building image analysis tools to assist those in the medical fields. We do not expect Physicians to totally understand everything in the paper.

Action and changes to the paper

Nonetheless, the manuscript has been thoroughly proofread (see the marked version).

Reviewer #3: 

1. Reviewer #3: -The paper should be interesting ;;;

-it is a good idea to add more photos of measurements, sensors + arrows/labels what is what (if any);;;

-figures should have high quality. ;;;;;

We thank the reviewer for this observation. Due to space and production quality, we cannot include all images, this is a continuing work, therefore we have published the tool and more images publicly on GitHub as follows

https://github.com/HeshamAlsaadi/COVNET45/tree/main/Research%20Images

Original Sources Link and README file

https://github.com/HeshamAlsaadi/COVNET45/blob/main/README.md

More images and diagrams will be added as soon as there are produced.

Action and changes to the paper

N/A

2. -What is the result of the analysis?;;

-what will society have from the paper?;;

-please add photos of the application of the proposed research, 2-3 photos ;;;

The tool will support scientists from all variety of fields, mainly IT developer who device tools for medical image analysis. COVNET45will allow them to harness the power and the use of AI technologies using several automatic intelligent techniques to process, detect and measure any giving object(s) and event(s) in raster image files highlighted in this paper with a discussion of microdroplet transmissions in airborne space. For example, biologist uses microscopic to study biological data to analyze cellular structure, organism, and characteristics. Understanding different cell characteristic under different conditions like germs or viruses, and effect of cell proliferation to examine the other conditions in-depth using AI algorithmic software’s. COVNET45 can examine anatomical cell and measure characteristics like capturing, tracking cells like microscopic studies a common tool as ImageJ. 

Action and changes to the paper

Please note that we have highlighted the results and contributions of this work in the abstract, introduction and new edited conclusions

3. -text should be formatted;;;;

-labels of figures should be bigger;;;;

We thank the reviewer for his comments, however, please note that the text is formatted as per the journal template. The same applies to figures captions, those are dictated by the journal style sheets.

Action and changes to the paper

N/A

4. -Is there a possibility to use the proposed research for other topics:

"Role of Hybrid Deep Neural Networks (HDNNs), Computed Tomography, and Chest X-rays for the Detection of COVID-19",;;;

"A Novel Method for COVID-19 Diagnosis Using Artificial Intelligence in Chest X-ray Images",;;;

We thank the reviewer for this comment. Yes, COVET45 may be used as a measurements tool to accurately examine anatomical cells and measure characteristics like capturing, tracking cells for microscopic studies. Also, it can examine analyze distance and scale based in world scale system. 

As for the X-ray images, COVNET45 is currently hosted in AWS and can be integrated with the AMAZON AI machine-learning platform. It is possible to extend the implementation of the block level representation of COVNET45 using hybrid deep neural network (HDNNs). Moreover, this can be implemented as a feature classification model in the tool if the model is trained with accurate and large dataset. 

In short, the model presented in the paper can be implemented; the trained model will allow us to detect any anomalies with precision and accuracy. However, this out of the scope of this publication for the time being.

Action and changes to the paper

N/A

5. -references should be from the web of science 2020-2022 (50% of all references, 30 references at least);;;

We thank the reviewer for his effort to ensure proper and high-quality sources are cited, we have developed and authored the work during 2020. That is why majority of the citation 15 out of 24 are from 2020 when the pandemic was at its peak. However, the nature of scholarly publication delays in submission, review and other factors may lead to citation looking old. 

Pls note that the reason behind this work was the limited reliable research that study microdroplets spread and transmissions. We tried our best to cite newly published work during 2021 and 2022 time focusing on web of science and other reputable venues.

Action 

The following papers have been discussed and cited.

14. HeshamAlsaadi, Aldwairi M. COVNET45: Artificial intelligence tool for the study of COVID-19 microdroplet spread across the human diameter and airborne spac; 2022. Available from: https://github.com/HeshamAlsaadi/COVNET45/blob/main/README.md.

15. HeshamAlsaadi, Aldwairi M. COVNET45 Images; 2022. Available from: https: //github.com/HeshamAlsaadi/COVNET45/tree/main/Research%20Images.

22. Yamamoto M, Kawamura A, ichi Tanabe S, Hori S. Predicting the infection probability distribution of airborne and droplet transmissions. Indoor and Built Environment. 0;0(0):1420326X221084869. doi:10.1177/1420326X221084869.

23. de Oliveira PM, Mesquita LCC, Gkantonas S, Giusti A, Mastorakos E. Evolution of spray and aerosol from respiratory releases: theoretical estimates for insight on viral transmission. Proceedings of the Royal Society A: Mathematical, Physical and Engineering Sciences. 2021;477(2245):20200584. doi:10.1098/rspa.2020.0584.

24. Rowe BR, Canosa A, Meslem A, Rowe F. Increased airborne transmission of COVID-19 with new variants, implications for health policies. Building and Environment. 2022;219:109132. doi:https://doi.org/10.1016/j.buildenv.2022.109132.

25. Niedrite L, Arnicans G, Solodovnikova D. Visualization of Indoor Sensor Data to Reduce the Risk of Covid-19 Infection. In: Doctoral Consortium/Forum@DB&IS; 2022.

6. -please compare advantages/disadvantages of other approaches;;;

We thank the reviewer for his useful comments, and we include below several paragraphs that deal with the advantages and disadvantages of the proposed methods in the literature.

Action 

New discussion added to related work

The following paragraphs deals with the comparison to related work

Yamamoto et al. proposed a probabilistic mathematical model for the distribution of airborne and droplets to predict Covid-19 infection [22]. For the parameters needed they relied on known estimation methods, however no real measurement or images processing was performed. It is true they applied this prediction model in a real office where they measured CO2 concentrations and the occupant positions, however there was no image analysis or measurement to study the spread of particles. Similarly, Oliveira et al. [23] presented a theoretical mathematical estimate of virus transmission to provide social distancing and ventilation measures, however none of that is comparable to actual microdroplet measurements capabilities of COVNET45.

More studies have dealt with the issue of airborne contamination paths, counting of pathogens, quantum concentration, carbon dioxide concentration and other measurements to give recommendations about ventilation and indoor capacity [24]. Additionally, there is a wealth of work in the area of Indoor air quality (IAQ) to use sensors and visualization in preventing indoor spread of SARS-CoV2 [25]. We believe those studies could benefit from the measurement capabilities of droplet transmission provided by COVNET45.

Existing discussion dealing with advantages and disadvantages of the proposed methods in the literature.

The research studies, mentioned earlier, adopted different techniques for scaling and measuring microdroplets infection transmissions either by simulating graphical representation of microscopic images or by visualizing the spread using Computational Fluid Dynamics (CFD) examination. Many of the discussed studies lack the evidentiary tools to validate their droplets measurement results, this makes it more complicated to authenticate the results in every research investigating the virus droplets spread through patient cough or sneeze. Some studies also supported only specific data formats

(CT scans, micrographic/microscopic images, CFD images, video graphics), and even when handling multiple formats, they cannot compare or validate results because of the restrictions presented by the investigation technology used. The fact that microdroplets patterns are ideally identifiable and recognizable makes it challenging to measure the precise size, even when using automated scaling systems implemented within the technology. Finally, extensive types of scales’ formats for accurately measuring microdroplets or any microscopic details within the hierarchy is challenging, which leaves the door open to investigate more optimized tools to explore and authenticate studies about the infectious disease.

-Conclusion: point out what have you done;;;;

We highlighted the paragraphs below in the conclusions that explain what we have done

Biologist uses microscopic images to study biological data to analyze cellular structure and organism characteristics. Artificial Intelligence algorithmic may assist in understanding different cell characteristic under different conditions such as germ, viruses and effect of cell proliferation. 

This paper presented an automatic measurement detection tool, dubbed ”COVNET45”. The tool provides five automatic intelligent techniques (SEGBON, EDGE, CED, DITHER and OPTIZ) to process, detect and measure any given objects and events in raster image files. COVNET45 will allow scientists to automatically optimize raster images, such as micrograph images of microscopic images, and perform precise measurements observation to any given object or particles. In addition, COVNET45 can examine anatomical cell and measure characteristics like capturing and tracking cells in microscopic studies.

---

## [Decision Letter · Decision Letter 2]

20 Mar 2023

Artificial intelligence tool for the study of COVID-19 microdroplet spread across the human diameter and airborne space

PONE-D-22-15591R2

Dear Dr. Aldwairi,

We’re pleased to inform you that your manuscript has been judged scientifically suitable for publication and will be formally accepted for publication once it meets all outstanding technical requirements.

Kind regards,

Anandakumar Haldorai, PhD

Academic Editor

PLOS ONE

Additional Editor Comments (optional):

Accepted

Reviewers' comments:

Reviewer's Responses to Questions

**Comments to the Author**

1. If the authors have adequately addressed your comments raised in a previous round of review and you feel that this manuscript is now acceptable for publication, you may indicate that here to bypass the “Comments to the Author” section, enter your conflict of interest statement in the “Confidential to Editor” section, and submit your "Accept" recommendation.

Reviewer #3: All comments have been addressed

2. Is the manuscript technically sound, and do the data support the conclusions?

Reviewer #3: Yes

3. Has the statistical analysis been performed appropriately and rigorously? 

Reviewer #3: N/A

4. Have the authors made all data underlying the findings in their manuscript fully available?

Reviewer #3: Yes

5. Is the manuscript presented in an intelligible fashion and written in standard English?

Reviewer #3: Yes

6. Review Comments to the Author

Reviewer #3: -----------------------------------------------------------------------------------------

7. PLOS authors have the option to publish the peer review history of their article (what does this mean?). If published, this will include your full peer review and any attached files.

Reviewer #3: No

---

## [Editor Report · Acceptance letter]

24 Mar 2023

PONE-D-22-15591R2 

Artificial intelligence tool for the study of COVID-19 microdroplet spread across the human diameter and airborne space 

Dear Dr. Aldwairi:

I'm pleased to inform you that your manuscript has been deemed suitable for publication in PLOS ONE. Congratulations! Your manuscript is now with our production department. 

Kind regards, 

on behalf of

Dr. Anandakumar Haldorai 

Academic Editor

PLOS ONE